# Behavioral responses of pyrethroid resistant and susceptible *Anopheles gambiae* mosquitoes to insecticide treated bed net

Maxwell G. Machani[1,2]*, Eric Ochomo[1], Fred Amimo[2], Wolfgang R. Mukabana[3,4], Andrew K. Githeko[1], Guiyun Yan[5], Yaw A. Afrane[6]*

1 Entomology Section, Centre for Global Health Research, Kenya Medical Research Institute, Kisumu, Kenya, 2 School of Health Sciences, Jaramogi Oginga Odinga University of Science and Technology, Kisumu, Kenya, 3 Department of Biology, Faculty of Science and Technology, University of Nairobi, Nairobi, Kenya, 4 Science for Health Society, Nairobi, Kenya, 5 Program in Public Health, College of Health Sciences, University of California, Irvine, CA, United States of America, 6 Department of Medical Microbiology, University of Ghana Medical School, College of Health Sciences, University of Ghana, Accra, Ghana

* machani.maxwe2011@gmail.com (MGM); yaw_afrane@yahoo.com (YAA)

**Data Availability Statement:** All relevant data used in the analysis are within the paper and its three Supporting Information files.

## Abstract

### Background

Long-lasting insecticidal nets are an effective tool in reducing malaria transmission. However, with increasing insecticide resistance little is known about how physiologically resistant malaria vectors behave around a human-occupied bed net, despite their importance in malaria transmission. We used the Mbita bednet trap to assess the host-seeking behavior of insecticide-resistant *Anopheles gambiae* mosquitoes under semi-field conditions. The trap incorporates a mosquito netting panel which acts as a mechanical barrier that prevents host-seeking mosquitoes from reaching the human host baiting the trap.

### Methods

Susceptible and pyrethroid-resistant colonies of female *Anopheles gambiae* mosquitoes aged 3–5 days old were used in this study. The laboratory-bred mosquitoes were color-marked with fluorescent powders and released inside a semi-field environment where a human subject slept inside a bednet trap erected in a traditional African hut. The netting panel inside the trap was either untreated (control) or deltamethrin-impregnated. The mosquitoes were released outside the hut. Only female mosquitoes were used. A window exit trap was installed on the hut to catch mosquitoes exiting the hut. A prokopack aspirator was used to collect indoor and outdoor resting mosquitoes. In addition, clay pots were placed outside the hut to collect outdoor resting mosquitoes. The F1 progeny of wild-caught mosquitoes were also used in these experiments.

### Results

The mean number of resistant mosquitoes trapped in the deltamethrin-impregnated bed net trap was higher (mean = 50.21± 3.7) compared to susceptible counterparts (mean + 22.4 ±

**Funding:** This study was supported by grants from the National Institute of Health (R01 A1123074, U19 AI129326, R01 AI050243, D43 TW001505). There was no additional external funding received for this study. The funders had no role in study design, data collection and analysis, decision to publish, or preparation of the manuscript.

**Competing interests:** The authors have declared that no competing interests exist.

1.31) (OR = 1.445; P<0.001). More susceptible mosquitoes were trapped in an untreated (mean = 51.9 ± 3.6) compared to a deltamethrin-treated bed net trap (mean = 22.4 ± 1.3) (OR = 2.65; P<0.001). Resistant mosquitoes were less likely to exit the house when a treated bed net was present compared to the susceptible mosquitoes. The number of susceptible mosquitoes caught resting outdoors (mean + 28.6 ± 2.22) when a treated bed net was hanged was higher than when untreated bednet was present inside the hut (mean = 4.6 ± 0.74). The susceptible females were 2.3 times more likely to stay outdoors away from the treated bed net (OR = 2.25; 95% CI = [1.7–2.9]; P<0.001).

## Conclusion

The results show that deltamethrin-treatment of netting panels inside the bednet trap did not alter the host-seeking behavior of insecticide-resistant female *An. gambiae* mosquitoes. On the contrary, susceptible females exited the hut and remained outdoors when a treated net was used. However, further investigations of the behavior of resistant mosquitoes under natural conditions should be undertaken to confirm these observations and improve the current intervention which are threatened by insecticide resistance and altered vector behavior.

## Introduction

Reduction in malaria morbidity and mortality over the past decade in sub-Saharan Africa is largely attributed to the effectiveness of long-lasting insecticidal nets (LLINs) [1]. This has been possible because the main malaria vectors primarily feed indoors at night, a behavioral pattern that coincides with the time when human hosts are indoors and asleep [2–4]. However, extensive use of insecticides has subjected mosquitoes to intensive selection pressure, resulting in the development of physiological and behavioral resistance [5]. To date, several studies have reported and described physiological resistance mechanisms of mosquitoes to insecticides with an aim of improving resistance management strategies [6–10]. However, behavioral resistance to insecticides is poorly documented despite its potential impact on the efficacy of vector control tools and its effect on residual malaria transmission [11].

The continued success of the current vector control interventions is dependent on the susceptibility of target mosquito populations to the insecticides used. Until recently, Pyrethroids were one of the insecticide classes advocated for vector control in public health due to their low mammalian toxicity, unique modes of action such as fast knockdown, excito-repellency effects and high insecticidal potency [12]. Some innovative nets treated with a combination of a pyrethroid and either a non-pyrethroid compound e.g. synergists (piperonyl butoxide) and pyriproxyfen are under investigation [13–15]. Most recently, some of these nets received conditional endorsement from WHO to be used in areas reporting moderate insecticide resistance to pyrethroids [16]. Over the past two decades, the use of insecticide-treated nets has increased, exerting greater selection pressure on malaria vector populations and resulting in higher incidences of pyrethroid insecticide resistance that is likely to affect the effectiveness of vector control [17]. Some studies have reported the spread of pyrethroid resistance and the mechanisms involved including target site insensitivity caused by kdr mutations [5, 18] and detoxification enzymes that metabolize the insecticide before reaching its target site [19]. However, it is less clear how the observed resistance affects current control measures.

Existing literature on behavioral changes associated with insecticide use comes mainly from pyrethroid susceptible mosquitoes but the data on the behavior of pyrethroid-resistant malaria vectors is sparse and, at times conflicting, highlighting the need for additional research. Insect behavioral avoidance response to insecticides can be referred to as the ability to move away from an insecticide-treated area without lethal consequences [20]. Two types of behavioral avoidance responses by mosquitoes have been largely recognized. These include irritancy (mosquitoes enter houses but leave early only after making physical contact with the treated surface) and excito-repellency (mosquitoes exit the treated area without making physical contact or after detecting insecticide vapour from a distance) [21]. The endophilic nature, the aggressiveness and the time vectors spend indoors, may have an impact on the effectiveness of residual insecticides as these traits determine the contact time with treated surfaces [22]. Increased foraging earlier in the evening or later in the morning, i.e. times when the human hosts are not protected by insecticide-treated bednets, has been observed within the principal African malaria vectors in the *An. gambiae* and *An. funestus* species complexes [3]. Exophily has also been observed as a consequence of indoor insecticide use [22, 23]. This switch in behavior may limit contact between aggressive susceptible malaria vectors and treated surfaces, hence threatening the efficiency of indoor interventions. However, with the increased use of insecticides indoors and the development of insecticide resistance, it is likely mosquitoes may not be able to avoid contact [24]. It is suggested that avoidance behavior in mosquitoes that have become insensitive to pyrethroids may weaken due to increased selection pressure exerted by the insecticides used [25]. Some authors assert that physiologically resistant mosquitoes may use the recognition of insecticides as a proxy for host presence [24, 26, 27]. It is unclear if mechanisms related to insecticide resistance may influence the behavior of malaria vectors, as any molecular change in the insect nervous system, may have a pleiotropic effect on nerve function and insect behavior [28].

Given the important role of the current vector control interventions as a means of reducing the burden of malaria transmission and increasing insecticide resistance, the behavior of physiologically resistant malaria vectors should be well defined. In this study semi-field experiments were performed to examine the behavior of the major malaria vector *Anopheles gambiae s.s.* (hereafter referred to as *An. gambiae*) towards a human-occupied Mbita bed net trap containing insecticide-treated or untreated netting panel. We hypothesized that pyrethroid-resistant mosquitoes seek and bite human hosts indoors in the presence of indoor-based vector control interventions. Unfed, susceptible mosquitoes leave the house through windows or eaves and seek blood meals elsewhere. This study provides information on how the behavior of physiologically resistant vectors may differ in comparison to their susceptible counterparts, an aspect that is poorly understood. Given the rapid development of insecticide resistance in a large number of malaria vectors, there is an urgent need for evidence-based studies on the behavior of malaria vectors in the presence of vector control interventions if the significant gains made in reducing malaria morbidity and mortality is to be maintained.

## Methods

### Mosquito strains used in the experiments

Mosquitoes used in this study consisted of a deltamethrin-selected resistant strain and an unselected strain of *An. gambiae* hereafter referred to as resistant and susceptible mosquitoes, respectively [29]. The mosquitoes were collected from Bungoma County in western Kenya. The colonies were selected and maintained at the Centre for Global Health Research, Kenya Medical Research Institute (KEMRI) in Kisumu, western Kenya, under standard rearing conditions of 27 ± 2°C temperature, relative humidity (RH) of 80 ± 10% and under a L12: D12 h

light: dark cycle. During the rearing process, each colonized strain had three independent lineages that started with 200–250 females at every new generation to limit bottleneck effects [29]. The progeny of F1 wild-caught mosquitoes from the same region were also used to undertake these experiments.

**Resistant strain.** This colony underwent deltamethrin selection after each generation. The 6th generation, which was used in this study was highly resistant with 20% mortality according to the WHO criteria [30]. According to Machani *et al*. [29], resistance in this colony was mainly mediated by the cytochrome P450 detoxification enzyme. The two *kdr* mutations 1014S and 1014F were present and at high frequencies.

**Susceptible strain.** This strain shared the same genetic background with the resistant colony but was reared in the absence of insecticide selection pressure. After nine generations without selection pressure, the population had almost lost resistance to deltamethrin (Mortality; 92%) and after 13 generations the population showed increased mortality (97.3%). The 14th generation was used in this study. The generation difference between the resistant and susceptible colony was due to the delayed development in the selected resistant colony.

**Wild population.** F1 progeny obtained from wild-caught *An*. *gambiae* female mosquitoes from Bungoma area where the resistant and susceptible colonies originated were also used in this study. Each female (mother) was identified by PCR as *An*. *gambiae s*.*s* according to the methods of Scott *et al*. [31]. The wild population had 56% resistance to deltamethrin. It is reported that the observed resistance was mediated by a mix of metabolic and *kdr* traits [18, 32–35].

## Semi-field set up

The study was carried out at the Centre for Global Health Research, Kenya Medical Research Institute, Kisumu, Kenya located near the equator in western Kenya. The release and recapture studies were conducted in an enclosed system dubbed the MalariaSphere. The system measured 20m long × 8m wide [36] with slanted roofing standing 3m high at the sides and 4.5m in the middle. The entire structure was covered with insect-proof screen netting that prevented mosquitoes inside the system from escaping into the environment, or vice versa (Fig 1A). The system was also double-doored for the same reason. Inside the system a 3m × 3m mud-walled hut was erected resembling a typical African village house with respect to size, design and mosquito exit/entry points (eaves, window and door) (Fig 1B). The MalariaSphere had local vegetation and grass growing in it to mimic the natural vegetation in the study area and to provide shelter for mosquitoes in the outdoor environment (Fig 1B). Two round clay pots were installed in the enclosure but outside the hut to act as outdoor mosquito resting sites (Fig 1C). A Mbita bednet trap with or without an insecticide-treated net panel was erected inside the hut (Fig 1D). Treated and untreated nets were used on different nights in the same hut. A human subject slept inside the Mbita bednet trap, treated or untreated, inside the hut each night. To offset any personal bias due to differential sleeping habits or relative attractiveness to mosquitoes, two sleepers were recruited for this experiment and took turns sleeping under the bed net. They were instructed not to consume alcohol or smoke and avoid deodorants during the study period. The volunteers who slept under the bed net trap served as bait to attract the mosquitoes into the hut but were not bitten because of the mechanical barrier provided by the netting panel.

## Mosquito host-seeking activity using Mbita bednet trap

The Mbita bed-net trap described by Mathenge *et al*. [37] was used to capture host-seeking mosquitoes. The trap is a modified conical bed net made of light white cotton cloth instead of mosquito netting fabric (Fig 2). The trap had two chambers, an upper trap chamber and a

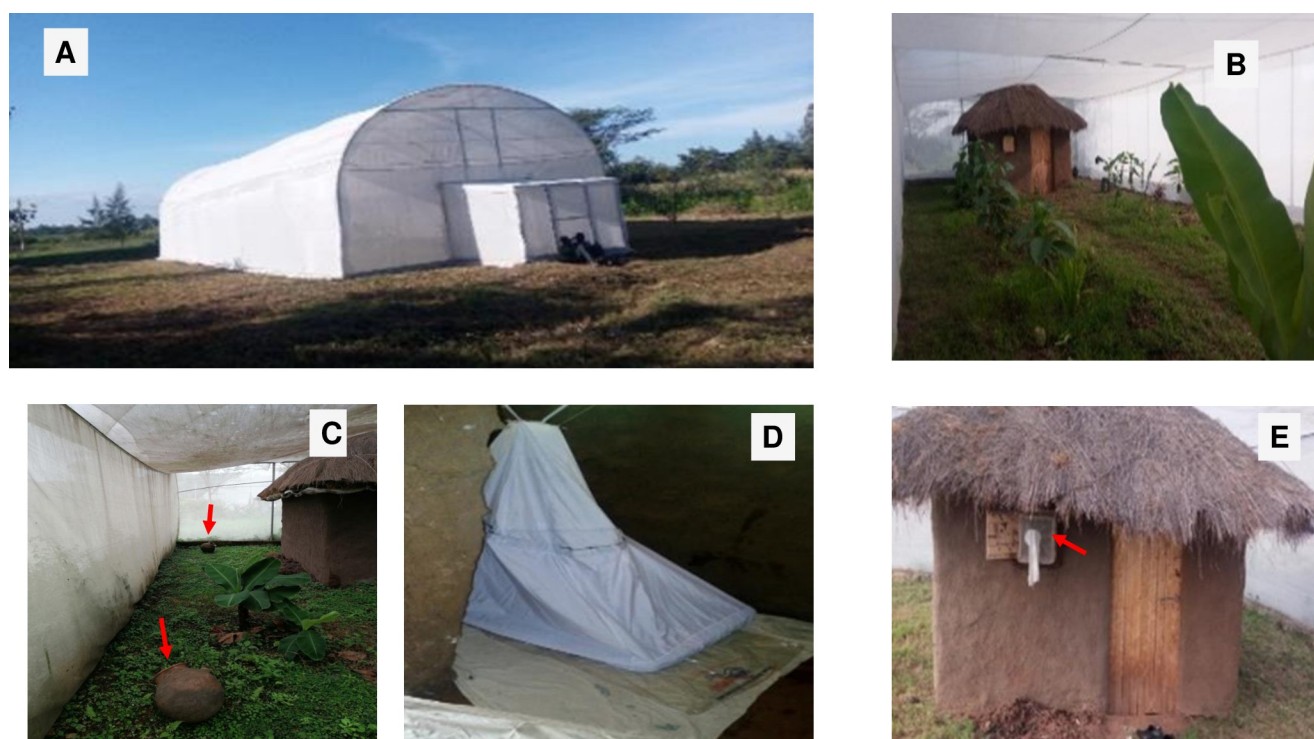

**Fig 1.** The semi-field set-up photographs showing (A) The screen house, (B) inside the screen house with a traditional hut and plants, (C) clay pots (pointed with red arrows) for collecting outdoor resting mosquitoes, (D) erected bed net trap (Mbita trap), (E) exit trap fitted on the window of the hut.

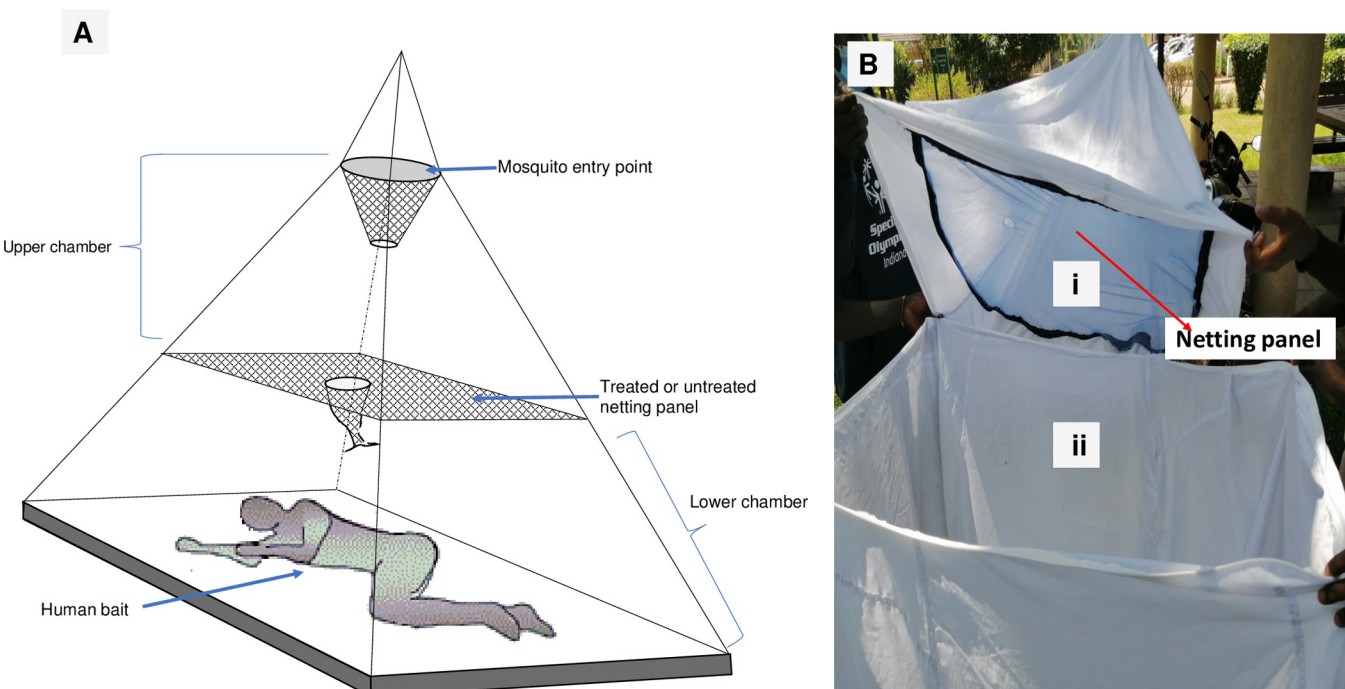

**Fig 2. The Mbita bed net trap.** Panel 'A' is an illustration of the trap with a person inside. Panel 'B' is a photograph of the trap showing (i) the upper chamber with a mosquito netting panel at its base and (ii) the lower chamber.

lower bait chamber, separated halfway by a netting panel (Fig 2A). The panel served to prevent host-seeking mosquitoes from reaching the human bait sleeping in the lower chamber (Fig 2B). For this experiment, the netting panels were either treated or untreated. The treated netting panels were cut from DawaPlus 2.0, a long-lasting insecticidal net (LLIN) containing 80 mg/m$^2$ deltamethrin. The working principle of the Mbita bednet trap is that host-seeking mosquitoes will respond to convective plumes, together with the accompanying body odor and exhaled breath from the human bait sleeping under the trap. The released mosquitoes after entering the house will fly up and down the trap responding to the mixed stimuli [37]. Some mosquitoes will follow and track the source of stimuli and end up being trapped inside the Mbita trap while others will choose not to follow the stimuli and stay outdoors.

The DawaPlus 2.0 nets were selected for this experiment based on the fact that they were distributed in the largest proportion in the study site by the National Malaria Control Programme in Kenya during the 2017 mass net campaign. The untreated netting panel inside the Mbita bed net trap was obtained from the local market in Kisumu, Kenya.

## Behavioural assay

Batches of 200 uninfected and unfed female *An. gambiae* mosquitoes aged 3–5 days old from the resistant or susceptible colonies were gently mouth-aspirated into a clean paper cup. The mosquitoes were sugar-starved for 6 hours before being released into the Malariasphere. The two strains were color-marked with either a green or pink fluorescent powder [FTX Series, Astral Pink; Swada (London) Ltd, London, U.K.] to distinguish them after simultaneous release into the semi-field environment. The powder was applied by filling a syringe (0.5 ml with 0.6 × 25 mm needle) with fluorescent powder. The syringe was held through the gauze at the top of the cup and in one gentle push, the powder was blown out of the syringe. This created a cloud of powder inside the cup with the mosquitoes [38]. To eliminate circadian effects resulting from environmental light: dark cycles, the colonies released were maintained in the laboratory under a fixed 12-hour light and 12-hour dark cycle. The release in the malariasphere was done early evening outside the hut and at the same time of the day (18.40 hrs) in all experiments. The volunteer entered the bednet trap 30 minutes after releasing the mosquitoes. Fifteen (15) tests were conducted with each net (treated or untreated net) for three months during the dry season. The release was done two times a week with a 3 days break to allow for the wash-out period. Windows of huts were fitted with exit traps to catch exiting mosquitoes (Fig 1E). The floor of the hut was covered with white sheets to ease the finding and collection of knocked-down mosquitoes. Host-seeking mosquitoes caught in the bed net trap were collected and recorded. The field population was used to validate the observed behaviors between these two strains because of any changes in behavior that may have resulted from colonization.

## Indoor and outdoor mosquito resting activity

Mosquitoes that were not caught in the bednet and window exit traps were collected from inside and outside the hut at 0700HRS using Prokopack aspirators (John W Hock, Gainesville, FL, USA). For mosquitoes resting indoors, walls and ceilings were systematically aspirated using progressive down and upward movements along its entire length. Collection of outdoor resting mosquitoes was done using clay pots (Fig 1C). To do this white mesh from a mosquito holding cage was placed over the mouth of the pot and mosquitoes resting inside the pot agitated, causing them to fly out of the pot into the cage [39]. The corners of the screen house and the vegetation cover were checked for the presence of resting mosquitoes using the Prokopack aspirator.

## WHO net bio-efficacy test

The insecticidal efficacy of the treated net was confirmed by exposing mosquitoes to DawaPlus 2.0 net for 3 mins according to the standard WHO cone bioassay procedure. This was done with 4–5 day old, non-blood fed, *An. gambiae s.s* mosquitoes. The bioassays included 5 replicates from both the insecticide-resistant and susceptible strains of *An.gambiae*. An average of five mosquitoes were placed per tube. The cone bioassays were conducted using DawaPlus 2.0 long-lasting insecticidal net treated with deltamethrin. The Kisumu strain and F1 progeny of wild-caught mosquitoes were also used in this experiment. After exposure, the groups of mosquitoes were placed in a single 1 L paper cup and provided with cotton wool soaked with 10% sugar solution for 24 hrs. Their knock-down status was measured 60 min post-exposure and mortality was recorded after 24 hrs. The survivors from the resistant colony were monitored for delayed mortality for an additional 48 hours. An untreated net was used as a negative control.

## Scientific and ethical clearance

This study was approved by the Ethical Review Board of the Kenya Medical Research Institute (KEMRI) protocol number SSC 3434. Prior to the commencement of the study, volunteers were given an information sheet describing study aims and procedures, risks and benefits of participating in the study. Written informed consent was obtained from individual volunteers before the experiments. The experiments were performed in accordance with the institution's guidelines and regulations.

## Statistical analysis

Data were entered into an Excel spreadsheet from where the distribution of vector collections was determined. The number of female mosquitoes caught in the bed net trap was interpreted as host-seeking mosquitoes. Mosquito house entry rate was calculated as the number of free mosquitoes collected indoors and those found inside bed net and exit traps divided by the total number released for each group.

Observations of host-seeking and exit behavior of insecticide-resistant and susceptible mosquito phenotypes were compared between treatments using a generalized linear model (GLM). A binomial distribution and logit link function were used to model the data. The effects of sampling nights on the number of mosquitoes trapped in the bed net trap was fitted as a random effect. The presence of insecticide on the netting panel inside the bed net trap and the number of mosquitoes released were fitted as fixed factors.

The insecticide-impregnated panel of mosquito netting present inside the bed net trap was considered bio-effective when the percentage of mosquitoes knocked down after 60 min post-exposure was above 95% or when 24-hour mortality or after 24 hours (delayed mortality) was above 80% in the WHO cone bioassays [40]. Statistical analysis was done using the statistical program Stata (Version 14, StataCorp, College Station, Texas).

## Results

### Responses of mosquitoes to bednet traps with untreated or insecticide-treated netting panels

We tested the response of resistant and susceptible mosquitoes to a human host sleeping under either insecticide-treated or untreated bed net traps placed inside the hut in the Malaria-Sphere. In 30 experimental nights (i.e. 15 treated and 15 untreated test repeats) out of 12,000 female *An. gambiae s.s* (resistant and susceptible) mosquitoes released, 55.5% (6663/12000)

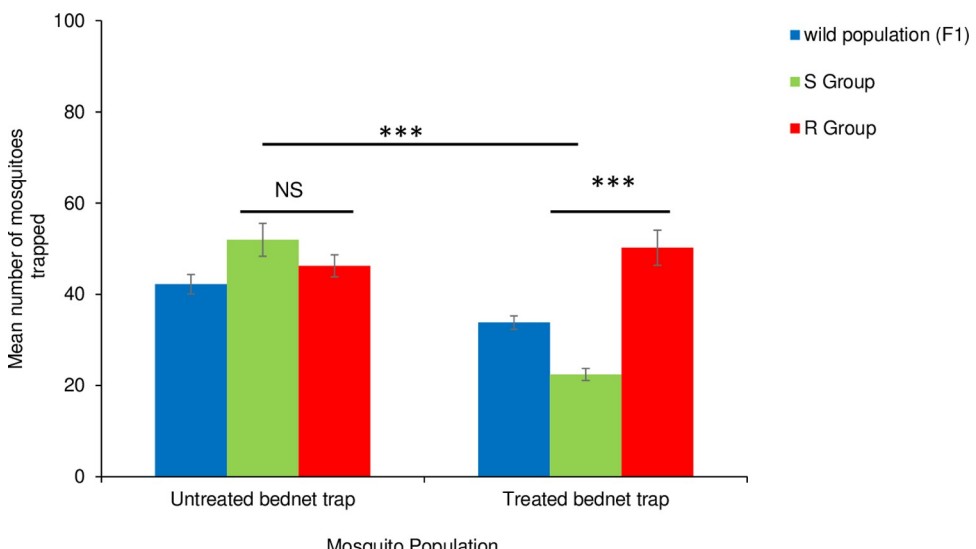

**Fig 3. Mean number of host-seeking mosquitoes from the three populations trapped in the treated and untreated Mbita bed net trap.** Error bars indicate the standard error of the mean.***, p<0.001, *NS* not significant.

were recovered (S1 Table). The mean number of resistant females trapped in the treated bed net trap was 50.21± 3.7 compared to that of the susceptible females 22.4 ± 1.31. The resistant females were more likely to seek a host sleeping under a treated bed net than susceptible mosquitoes (OR = 1.445; 95% CI = [1.25–1.68]; P<0.0001, Fig 3). Significantly more susceptible females were trapped in an untreated (mean: 51.9 ± 3.6) than a treated bed net trap (mean: 22.4 ± 1.3). When the untreated net was present the susceptible mosquitoes were 2.7 times more likely to search for a host than when a treated bed net was present (OR = 2.65; 95% CI = [2.29–3.05]; P<0.0001, Fig 3). GLM analysis indicated that there was a significant effect of treatment on the number of mosquitoes trapped, with more mosquitoes being caught in the untreated versus treated bed net trap (S2 Table).

For the wild population, a total of 1013 (50.6%) mosquitoes were recaptured out of 2000 F1 females released. The proportion of the wild population caught in the untreated bed net trap was slightly higher 41.4% (211/509) % compared to treated bed net trap 33.8% (169/504) (Fig 3). However, this was not statistically significant (OR = 0.773; P = 0.489).

The mortality of the resistant population trapped in the treated bed net trap was 77.7% (549/706) and 85.2% (144/169) for the wild population. All the susceptible mosquitoes trapped in the insecticide-treated bed net died.

## Insecticide induced exophily of resistant and susceptible populations

Overall, the proportion of mosquitoes that entered the hut when the treated net was present was high 51.1% (95%, CI = [49.3–52.9]) for the resistant than the susceptible strain 39.8.1% (95%, CI = [38.1–41.5]) of *An. gambiae*. The proportion of susceptible mosquitoes entering the hut increased to 52.6% (95%, CI = [50.8–54.4]) when the untreated net was present (Table 1). The number of susceptible females caught exiting the hut when a treated bed net trap was present was 22.3 ± 2.9 compared to the resistant females (Mean: 4.2 ± 0.8). The resistant females were less likely to exit the house when a treated net was present compared to the susceptible females (GLM, OR = 0.54; P<0.0001). When the untreated bed net was present, the number of mosquitoes exiting reduced for the susceptible group (Mean: 2.4 ± 0.8)

**Table 1. Number of mosquitoes recaptured and proportion of mosquitoes entering and exiting the hut following the use of treated and untreated bednet trap.**

| Status of Bednet trap | Mosquito population | No. released | No. recaptured | hut entry (%), 95%Cl | No. Exiting (Mean± SEM |
|---|---|---|---|---|---|
| Treated | Resistant | 3000 | 1642 | 51.1[49.3–52.9] | 4.6 ± 0.80 |
| | Susceptible | 3000 | 1745 | 39.8[38.1–41.5] | 22.3 ± 2.90 |
| Untreated | Resistant | 3000 | 1628 | 52.1[50.3–53.5] | 2.3 ± 0.70 |
| | Susceptible | 3000 | 1648 | 52.6[50.8–54.4] | 2.4 ± 0.80 |
| Treated | Wild population (F1) | 1000 | 504 | 86.7[83.7–89.7] | 16 ± 2.12 |
| Untreated | Wild population (F1) | 1000 | 509 | 92.5[90.2–94.8] | 3.6 ± 0.51 |

(Table 1). Overall, the susceptible females were 4.6-fold more likely to exit the house when treated bed net trap was present than when the bed net was untreated (GLM, OR = 4.64; 95% CI = [3.3–6.5]; P<0.0001). For the wild field population, 16 ± 2.1 of the recovered mosquitoes were caught in the exit trap when the treated bed net trap was present, while 3.6 ± 0.5 when the untreated net was used.

## Mosquito indoor versus outdoor resting behavior in relation to insecticide use

The average number of mosquitoes caught resting inside the hut when a host slept under a treated bed net trap was higher 26.4 ± 2.33 for resistant females compared to susceptible females 18.1 ± 1.34. There was no difference between the proportion of resistant and susceptible female mosquitoes caught resting indoors in the presence of an untreated bed net trap (OR = 1.1; 95% CI = [0.97–1.28]; P = 0.121) (Fig 4). The number of susceptible females caught resting outside the hut when a treated net trap was used, was higher 28.6 ± 2.22 compared to when an untreated net was present 4.6 ± 0.74. The susceptible mosquitoes were 2.3 times more likely to stay outdoors away from the treated bed net (OR = 2.25; 95% CI = [1.7–2.9]; P<0.0001; Fig 4).

For the wild population, the average number of females caught resting inside or outside the hut when a treated bed net was present was 37.6 ± 2.4 versus 13.4 ± 2.34, respectively, compared to the untreated bed net trap (Mean; 47.4 ± 2.1 vs 8.6 ± 1.03 respectively, Fig 4). Even though the proportion resting indoor or outdoor was high when the treated bed net was present, there was no significant difference (Indoor: OR = 1.2; 95% CI = [0.94–1.54]; P = 0.139; outdoor: OR = 1.11; 95% CI = [0.72–1.71]).

## LLIN bioassay and knockdown rates against resistant and susceptible colonies

Prior to the semi-field trials, the efficacy of the treated bed net was evaluated (S3 Table). The knockdown response of the resistant females exposed to DawaPlus 2.0 for 60 minutes was 7% whilst 83% of the susceptible population was knocked down. The 24-hour mortality rate for the resistant colony was 13% (95% CI = [9.1–15.9]) whilst 92% (95% CI = [89.4–94.9]) for susceptible population (Fig 5A). The knockdown rate for F1 progeny of the wild population when exposed to DawaPlus 2.0 for 60 minutes was 36% while the mortality rate after 24 hours was 59% (95% CI = [50.3–67.9]). Kisumu reference susceptible strain had a knockdown rate of 96% and a 100% mortality rate when exposed to DawaPlus 2.0. The mortality rate between 24 and 72 hours (within 1 and 3 days) after last exposure of resistant females to DawaPlus 2.0 ranged from 13% (95% CI = [9.1–15.9] to 16.4% (95% Cl = [12.6–20.2]) (Fig 5B).

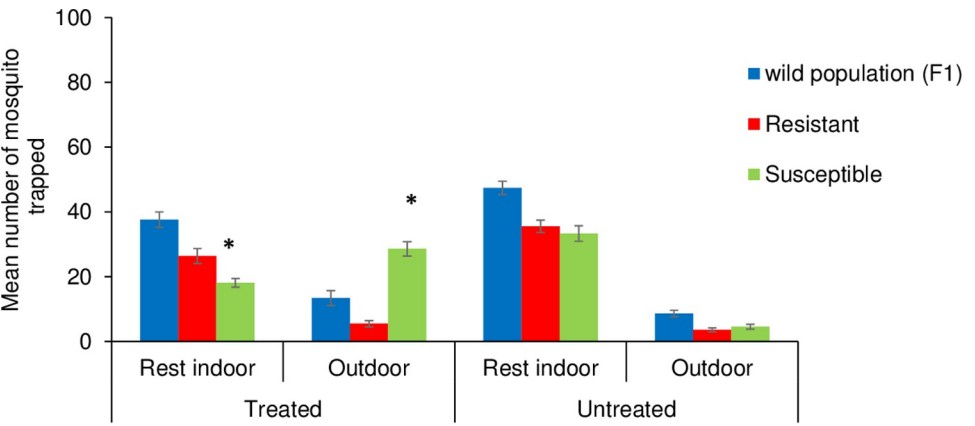

Treatment

**Fig 4. Mean number of mosquitoes resting indoors and outdoors when a treated and untreated bed net trap was present.** Bars labeled with asterisks* indicate findings that are significantly different from others when a treated and untreated bednet is used. Error bars indicate the standard error of the mean.

## Discussion

Physiological resistance in mosquito populations to common public health insecticides across Africa is widely reported [17, 41]. However, the knowledge of behavioral responses associated with resistance and downstream impact and efficacy of LLINs is scarcely documented [42]. Monitoring the host-seeking behavior of physiologically resistant mosquitoes in the presence of indoor vector control tools is necessary to determine whether the efficacy of the tools could be compromised with the resistance profiles or whether they can be optimized. This study provides insights into the behavior of pyrethroid-resistant *An. gambiae* when they encounter pyrethroid-based LLINs in a free-flight environment similar to the field settings. The results demonstrate that in the presence of a treated net, the host-seeking performance was not altered for resistant females, unlike the susceptible females that were observed to exit the house and remained outdoors when a treated net was used.

One of the consequences of the massive roll-out of LLINs is the change in mosquito behavior where the interventions may select vectors with increased exophily (feeding outdoors early in the evening or morning hours when LLINs are not in use) because of the exposure to insecticides [11]. This study observed a large proportion of host-seeking susceptible females exiting the house and resting outdoors than resistant females when the treated net was present. The observed behavior confirms the excito repellency effect of pyrethroid-treated nets, suggesting that susceptible mosquitoes may be pushed from indoor-treated environments and resort to search blood meals outdoors or rest outdoors and initiate their search for a host soon after dusk, leading to increased outdoor transmission. On the other hand, the findings suggest, physiologically resistant malaria vectors that have developed the capacity of blood-feeding or resting indoors in the presence of LLINs, may compromise the effectiveness of LLINs, maintaining the indoor malaria transmission. The current findings emphasize the need for continuous monitoring and designing of novel resistance management strategies as the prevalence and intensity of resistance at different locations may have an effect on malaria transmission [43].

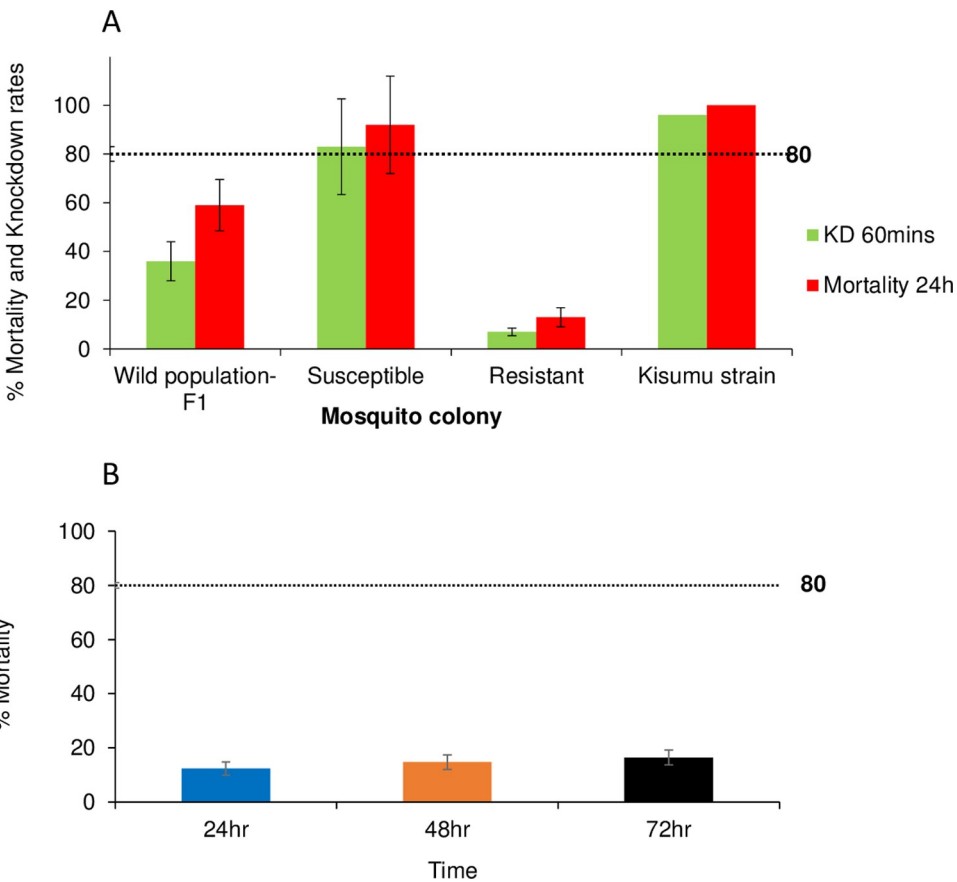

**Fig 5.** Percentage of A) knockdown and mortality rates of the three populations of *An.gambiae* (F1 from wild population, susceptible strain, resistant Strain) exposed to insecticide-treated nets (DawaPlus 2) in WHO net bioassay test for 3 minutes. Knock-down was measured after 1h and mortality after 24h, B) 72 hour delayed mortality for the resistant strain. Kisumu strain is the standard WHO susceptible reference population. Error bars indicate 95% confidence intervals. The 80% mortality threshold for calling full susceptibility based on the WHO criteria is indicated.

Examples of spatial avoidance have been observed in malaria vectors in the field, displaying increased outdoor host-seeking and resting outdoors following the implementation of IRS and ITNs [44, 45]. When F1 progeny of wild *Anopheles gambiae s.l* were released, the proportion trapped attempting to bite and exiting the hut was slightly high when the untreated net was present compared to when the treated net was in place although these findings were not statistically significant. It is noteworthy to mention that the wild population originated from the same region and shared the same background as the resistant and susceptible strain. Previous studies observed that both *kdr* and metabolic resistance drove pyrethroid resistance in this mosquito population. The *kdr* east (1014S) mutation was reported at high frequencies, unlike 1014F which was at a low frequency [18, 34, 46]. The difference in behaviours between the resistant and the F1 wild population could be due to the heterogeneity of the field population in terms of their response to the insecticide. This indicates that a substantial part of residual malaria transmission is occurring outdoors, raising the questions on the effectiveness of LLINs in reducing malaria infections when susceptible indoor feeding mosquitoes are diverted to feed outdoors when people are outside LLINs.

The strategy of LLINs in malaria prevention is to deter mosquitoes from entering houses and to reduce blood-feeding rates, both of which are achieved as a consequence of excito-

repellent and killing effects of the pyrethroids [47]. In this study, a higher proportion of the resistant females were caught in the treated bed net trap compared to the susceptible females. The WHO net bioassay tests confirmed lower mortality of resistant mosquitoes suggesting that nets were effective towards susceptible mosquitoes. One plausible explanation for the difference in host-seeking behaviour is the pleiotropic effects on nerve function associated with a point mutation in the voltage-gated sodium channels of resistant mosquitoes, as it interferes with the sensitivity of the sensory nervous system to pyrethroids resulting in reduced avoidance behavior [48, 49]. Studies carried out by Diop *et al.* [50] in the laboratory on host-seeking behavior of mosquitoes in the presence of damaged treated nets using a wind tunnel, observed increased performance of resistant females with 1014F *kdr* mutations compared to susceptible. This implies that in the field, physiologically resistant mosquitoes are likely to spend more time in search of a host in the presence of insecticides increasing their probability of encountering a host, unlike their susceptible counterparts that could either die after contacting the insecticides or repelled from indoor dwellings. In nature, pyrethroid-resistant mosquitoes have been found resting inside holed LLINs [51]. Such behavior may compromise the efficacy of the current indoor-based vector control tools resulting in increases in malaria transmission indoors [4]. Recent studies from western Kenya observed high resistance levels, rates of human blood index and sporozoite rates in the mosquitoes resting indoors compared to the mosquitoes collected resting outdoors [34, 52]. The study findings are in agreement with similar studies that have observed reduced host-seeking performance of susceptible mosquitoes in the presence of LLINs unlike the resistant mosquitoes whose behavior was not altered [26, 51, 53, 54].

This study had limitations, based on genotyping results the *kdr* mutations(1014S) was detected at high frequency in our phenotypically susceptible mosquitoes as the mutation was already fixed in the parent population [29], raising questions about the effect of 1014S mutation on the behaviour of this population in the presence of pyrethroids. Although the 1014S mutation associated with pyrethroid resistance was observed in the susceptible colony, the phenotypically resistant mosquitoes had both 1014S and 1014F *kdr* mutations at high frequencies and increased monooxygenase enzymes. Also due to the design of the trap, which has a funnel-shaped entry point with no return port for mosquitoes trapped, it's difficult to measure the response of mosquitoes after contacting the treated net, however, the catches will likely reflect the true composition of the host-seeking mosquito population.

The findings of this study show that despite the coverage of the indoor interventions, it is evident that not all malaria transmission can be controlled with the existing tools that are indoor-based. The population of vectors that move outdoors are not taken care of, a situation that creates a pressing need for supplementary vector control tools to control residual transmission.

## Conclusion

The results show that in the presence of a pyrethroid treated net, the host-seeking performance was not altered for the resistant mosquitoes, unlike the susceptible females that were observed to exit the house and remain outdoors when a treated net was used. This might be a reason for continued malaria transmission indoors in areas with high pyrethroid resistance despite the scaling up of vector control interventions and increased outdoor malaria transmission in sub-Saharan Africa. This situation calls for urgent deployment of control tools that can complement the current vector control methods to tackle outdoor malaria transmission. However, further investigations of the behavior of resistant mosquitoes under natural conditions should be undertaken to confirm these observations and improve the current interventions which are threatened by insecticide resistance and altered vector behavior.

## Supporting information

**S1 Table. Results on the fate of all released mosquitoes of each strain when an insecticide treated and untreated panel was present.**
(DOCX)

**S2 Table. Results from the mixed linear model fit by maximum likelihood examining the impact of bednet status and number of mosquitoes released, while considering night as a random effect, on the number of mosquitoes trapped during the night.**
(DOCX)

**S3 Table. Summary results on Bioefficacy of deltamethrin treated against pyrethroid resistant and susceptible *Anopheles gambiae* mosquitoes.**
(DOCX)

## Acknowledgments

The authors wish to thank the volunteers for their participation in this study. We acknowledge Kevin Owour, Joyce Osoro and the Entomology Laboratory team at the Kenya Medical Research Institute, Kisumu, for providing technical support. The permission to publish this study was granted by the director of the Kenya Medical Research Institute.

## Author Contributions

**Conceptualization:** Maxwell G. Machani, Eric Ochomo, Guiyun Yan, Yaw A. Afrane.

**Data curation:** Maxwell G. Machani.

**Formal analysis:** Maxwell G. Machani.

**Funding acquisition:** Guiyun Yan, Yaw A. Afrane.

**Investigation:** Maxwell G. Machani, Eric Ochomo, Yaw A. Afrane.

**Methodology:** Maxwell G. Machani, Eric Ochomo, Andrew K. Githeko, Guiyun Yan, Yaw A. Afrane.

**Supervision:** Eric Ochomo, Fred Amimo, Wolfgang R. Mukabana, Andrew K. Githeko, Guiyun Yan, Yaw A. Afrane.

**Validation:** Eric Ochomo, Fred Amimo, Wolfgang R. Mukabana, Andrew K. Githeko, Guiyun Yan, Yaw A. Afrane.

**Writing – original draft:** Maxwell G. Machani.

**Writing – review & editing:** Eric Ochomo, Fred Amimo, Wolfgang R. Mukabana, Andrew K. Githeko, Guiyun Yan, Yaw A. Afrane.

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
