## [Decision Letter · Decision Letter 0]

26 Jan 2022

PONE-D-21-36181Behavioral responses of pyrethroid resistant and susceptible Anopheles gambiae mosquitoes to insecticide treated bed netPLOS ONE

Dear Dr. Afrane,

Thank you for submitting your manuscript to PLOS ONE. After careful consideration, we feel that it has merit but does not fully meet PLOS ONE’s publication criteria as it currently stands. Therefore, we invite you to submit a revised version of the manuscript that addresses the points raised during the review process.

Note that while both reviewers felt your work has a relevant contribution, they suggested modifications that should be adequately answered and eventually addressed in a revised manuscript. Please give special attention to Rev #1 concern on the limitation of the collection mode. Note that both reviewers coincided on the need to review typos and grammatical errors carefully.

We look forward to receiving your revised manuscript.

Kind regards,

Pedro L. Oliveira

Academic Editor

PLOS ONE

Journal Requirements:

“This study was supported by grants from the National Institute of Health (R01 A1123074, U19 AI129326, R01 AI050243, D43 TW001505). There was no additional external funding received for this study. The funders had no role in study design, data collection and analysis, decision to publish, or preparation of the manuscript.”

Please note that funding information should not appear in other areas of your manuscript. We will only publish funding information present in the Funding Statement section of the online submission form.

“This study was supported by grants from the National Institute of Health (R01 A1123074, U19 AI129326, R01 AI050243, D43 TW001505). There was no additional external funding received for this study. The funders had no role in study design, data collection and analysis, decision to publish, or preparation of the manuscript.”

4. Please review your reference list to ensure that it is complete and correct. If you have cited papers that have been retracted, please include the rationale for doing so in the manuscript text, or remove these references and replace them with relevant current references. Any changes to the reference list should be mentioned in the rebuttal letter that accompanies your revised manuscript. If you need to cite a retracted article, indicate the article’s retracted status in the References list and also include a citation and full reference for the retraction notice

Reviewers' comments:

Reviewer's Responses to Questions

**Comments to the Author**

1. Is the manuscript technically sound, and do the data support the conclusions?

Reviewer #1: No

Reviewer #2: Yes

2. Has the statistical analysis been performed appropriately and rigorously? 

Reviewer #1: I Don't Know

Reviewer #2: Yes

3. Have the authors made all data underlying the findings in their manuscript fully available?

Reviewer #1: Yes

Reviewer #2: Yes

4. Is the manuscript presented in an intelligible fashion and written in standard English?

Reviewer #1: No

Reviewer #2: Yes

5. Review Comments to the Author

Reviewer #1: The paper has an interesting question, very important to malaria control and surveillance, but the authors based their behavioral conclusions on weak data.

Would you please take a careful look at the text regarding spelling, formatting, and grammar issues?

Introduction

There are some minor conceptual errors, which should be corrected.

Line 79: please substitute "period" with "phase."

Material and Methods

It would be interesting to describe the protocol used to select the colonies for deltamethrin. Alternatively, it would help if you mentioned it is the same colony studied in Machani et al., 2020.

Did you also check for mutations in the susceptible strain after the 14th generation?

Were the wild-type population and the population used to form the resistant and susceptible colonies collected in the same area? I mean, do you think They have similar genetic backgrounds?

Lines 213-214: "Mosquito releases were done outside the hut and at the same time of day (1840hrs) to avoid circadian effects." Can you describe the environment outside MalariaSphere at this time? Depending on the month, at this time, there was sunlight, or it was twilight or even early evening. So maybe you didn't avoid circadian effects that much.

Lines 221- 222: I didn't understand this validation.

Results

Figures 3-5 and Table 1: please indicate in the figures/table the groups that presented statistical differences.

The results presented in figure 5 are poorly described in the text. The Kisumu strain is mentioned only in the figure legend, and the colors used in figure 5B are confusing. You could present data in a more straightforward manner.

Discussion

It is not clear if the authors could answer the paper's central question once the conclusions are based only on the collection of mosquitoes after a particular time. It is difficult to conclude which behavior was more affected in the experiments, the irritancy of the excito-repellency as there are no images inside the huts.

Another interesting point is about the wild-type population. The WT pop exits more than the resistant pop when bednets are treated. Was it expected or not? It is necessary to discuss this data, mainly because it is not clear what is the genetic background of this population concerning kdr mutations.

Reviewer #2: This manuscript by Yaw Afrane et al describes a very nice experiment examining the effect of insecticide resistance on behavioural responses to insecticide treated and untreated nets, in a semi-field environment. As the authors mention, this is an area that has not been very well explored but is important to understanding how interventions work in areas of resistance, highlighting remaining transmission risk, and to designing better interventions. The results are interesting, clearly described, and well interpreted, and well framed by an informative and targeted Introduction and clear Discussion.

I just have a few minor suggestions.

The abstract does not mention the experiment done with F1 wild mosquitoes.

Lines 87-91 – this section is a bit out of date, ignoring the next generation ITNs which have already been deployed in many sites, for example Royal Guard and Interceptor G2 which contain novel chemistries.

Lines 152-153 – how was the relative contribution of resistance mechanisms determined, to be able to state that resistance was ‘mainly mediated’ by P450s?

Lines 199-201 – this section is not very clear, and it would be good to expand it to describe the behaviour of mosquitoes in more detail.

What is the ‘green dye’ that was used to mark mosquitoes? Fluorescent dust? Give details, with supplier.

Lines 221-222 – it is not clear to me how using F1 wild mosquitoes ‘validates’ these behaviours. Please explain this more clearly.

Statistical analysis section – would be clearer if it was divided into separate paragraphs.

Line 285 – were these 2,000 F1 mosquitoes included in the 12,000 mentioned in the previous paragraph?

Lines 288-291 – should be a separate paragraph

The figures are nice and clear, and I appreciate the use of photos and the labelled diagram of the Mbita bednet trap. I would like to suggest a figure is added, perhaps in place of Figures 3 and 4 – a stacked bar graph, to show the fate of all released mosquitoes of each strain with treated and untreated traps, would show all data from the experiment clearly in one place for easy comparison.

The first paragraph of the Discussion is quite out of date. Reference 5 is from 2011, and since then there has been a fair amount of research into the effects of insecticide resistance both on the impact of ITNs and on the sub-lethal effects of pyrethroid exposure on resistant mosquitoes, by the same authors and others. It is true that there has been less investigation of the effects on behaviour of mosquitoes, see Review and Meta-Analysis of the Evidence for Choosing between Specific Pyrethroids for Programmatic Purposes by Lissenden et al 2021. However, Phillip McCall at the Liverpool School of Tropical Medicine has done some work on behavioural responses that deserves mention here, albeit in a lab setting. The final sentence of this paragraph is an important new observation.

Lines 345-361 – are the authors able to comment on how these two effects (more exit of susceptible mosquitoes v more biting by resistant mosquitoes) might balance against each other in their effect on malaria transmission and/or efficacy of ITNs? At least they could comment that the prevalence and intensity of resistance in a given area would affect this balance at a local level.

Lines 362-364 – the authors describe ITN’s effect giving personal protection, but the other way that ITNs work is to provide community protection by killing mosquitoes, and this should be mentioned.

Lines 382-384 – the fact that the ‘susceptible’ strain carried the kdr mutation is an interesting observation, suggesting that this resistance mechanism is not involved in altered behaviour in resistant mosquitoes, but that others might be (such as point-mutations mentioned in the previous paragraph). It might be interesting to expand this sentence to discuss this.

In the Conclusion the effect of outdoor biting is discussed, but there will also, I presume, be more indoor biting by resistant mosquitoes which are not deterred from blood feeding.

The manuscript needs a careful edit to remove typos, formatting errors, and some awkward sentence structures and inaccurate wording.

6. PLOS authors have the option to publish the peer review history of their article (what does this mean?). If published, this will include your full peer review and any attached files.

Reviewer #1: No

Reviewer #2: No

---

## [Author Response · Author response to Decision Letter 0]

16 Mar 2022

Response to reviewer comments

The following are our point-by-point responses to the comments raised by the reviewers. We thank the reviewers for their constructive criticism and insights that have helped to improve this paper.

Comments Responses

 Reviewer # 1

 The paper has an interesting question, very important to malaria control and surveillance, but the authors based their behavioral conclusions on weak data.

Would you please take a careful look at the text regarding spelling, formatting, and grammar issues? Response: We thank the reviewer for the comments raised. We have carefully revised the manuscript giving much attention to the grammatical and typographical errors.

Comment: There are some minor conceptual errors, which should be corrected. Line 79: please substitute "period" with "phase." Response: We have addressed the conceptual errors and also replaced “period” with “time” (Line 80) as suggested by the reviewer.

Comment: Material and Methods It would be interesting to describe the protocol used to select the colonies for deltamethrin. Alternatively, it would help if you mentioned it is the same colony studied in Machani et al., 2020. Response: We have cited Machani et al 2020 (REF. 29) in Line 151-152 to show that these were the same mosquito colonies studied by the authors.

Comment: Did you also check for mutations in the susceptible strain after the 14th generation? Response: We checked the mutations in the 13th generation of the susceptible strain before using the mosquitoes in our experiments in the 14th generation. In the methodology section, we mentioned (Line 161) that we only released the 14th generation of the susceptible strain.

Comment: Were the wild-type population and the population used to form the resistant and susceptible colonies collected in the same area? I mean, do you think They have similar genetic backgrounds? Response: The wild-type population used were collected from the same region as the two strains (resistant and susceptible) and also share the same genetic background. We mentioned this in the methodology section Line 151-152, 166.

Comment: Lines 213-214: "Mosquito releases were done outside the hut and at the same time of day (1840hrs) to avoid circadian effects." Can you describe the environment outside MalariaSphere at this time? Depending on the month, at this time, there was sunlight, or it was twilight or even early evening. So maybe you didn't avoid

circadian effects that much. Response: The colonies used in this study were maintained in the laboratory under a fixed 12-hour light and 12-hour dark cycle . In addition, the mosquitoes were released at 18.40 hours in all experiments, so eliminating any circadian effects that might have influenced the results (Line 221-223). Furthermore, the malariasphere in which the experiments were done is located near the equator (Line 172) where the lengths of the day and night are similar meaning that there were minimal differences in environmental conditions outside the malariasphere.

Comment: Lines 221- 222: I didn't understand this validation. Response: The resistant and susceptible strains used in this work had been colonized for 6 and 14 generations, respectively. The field population was used to validate the observed behaviors between these two strains because of any changes in behavior that may have resulted from colonization (Line 230-232).

Comment: Figures 3-5 and Table 1: please indicate in the figures/table the groups that presented statistical differences. Response: We have shown in the figures and tables the groups that were statistically significant.

Comment: The results presented in figure 5 are poorly described in the text. The Kisumu strain is mentioned only in the figure legend, and the colors used in figure 5B are confusing. You could present data in a more straightforward manner. Response: This section has been revised from Line 337-346. Kisumu reference susceptible strain has also been included in the text. We have changed the colors for Figure 5B to avoid confusion. We thank you for figuring this error out.

Comment: It is not clear if the authors could answer the paper's central question once the conclusions are based only on the collection of mosquitoes after a particular time. It is difficult to conclude which behavior was more affected in the experiments, the irritancy of the excito-repellency as there are no images inside the huts. Responses: We strongly feel that the findings presented in our study answer the questions that we sought to investigate. The aim of this study was to investigate if pyrethroid-resistant mosquitoes could seek and bite human hosts indoors despite the presence of indoor-based vector control interventions. Otherwise, the mosquitoes would deliberately escape the treated environment and bite outdoors or rest away from the treated environment. We compared these behavioural responses with the susceptible counterparts and from our findings we observed that the host-seeking behavior of resistant mosquitoes was not altered unlike the susceptible ones when treated nets were present. The susceptible mosquitoes deliberately escaped the treated environment and a higher proportion was observed resting outdoors when the treated net was present than when the untreated net was used.

Indeed, we did not have images in the hut to record the irritancy because logistically it was not possible and from the design of the study we could not measure irritancy as there was no direct contact, the treated part was a panel inside the Mbita trap and only the host-seeking mosquitoes were caught inside. The catches in the Mbita trap most likely reflect the true composition of the host-seeking mosquito population. Also, the study assumed the proportion caught exiting the hut was due to excito- repellency of insecticides used on LLIN panel.

Comment: Another interesting point is about the wild-type population. The WT pop exits more than the resistant pop when bednets are treated. Was it expected or not? It is necessary to discuss this data, mainly because it is not clear what is the genetic background of this population concerning kdr mutations. Response: The wild-type population shared the same origin as the resistant and susceptible populations. We have included detailed information Line 376-385 in the text about the wild population. We expected to see a difference between the resistant and wild-type population because the latter population was heterogeneous in terms of response to the insecticide as we have both susceptible and resistant individuals in this population. The high frequency of 1014F mutations could have also contributed to the observed difference as the mutation was high in the selected resistant population than the wild-type population.

 Reviewer #2

General Comment: This manuscript by Yaw Afrane et al describes a very nice experiment examining the effect of insecticide resistance on behavioural responses to insecticide-treated and untreated nets, in a semi-field environment. As the authors mention, this is an area that has not been very well explored but is important to understanding how interventions work in areas of resistance, highlighting remaining transmission risk, and to designing better interventions. The results are

interesting, clearly described, and well interpreted, and well framed by an informative and targeted Introduction and clear Discussion.

I just have a few minor suggestions. Response: We thank the reviewer for appreciating our work

Comment: The abstract does not mention the experiment done with F1 wild mosquitoes

 Response: We have included a statement in the abstract section Line 54 indicating that “The F1 progeny of wild-caught mosquitoes were also used in these experiments”. 

Comment: Lines 87-91 – this section is a bit out of date, ignoring the next generation ITNs which have already been deployed in many sites, for example Royal Guard and Interceptor G2 which contain novel chemistries. Response: We have included a statement indicating the assessment of novel chemistries Line 92-94 and the recent conditional deployment of the next generation nets Line 94-95. We have cited Ngufor et al 2020 on the efficacy of Royal Guard and WHO report 2017 on the conditional deployment of next-generation nets (PBO nets).

“Some innovative nets treated with a combination of a pyrethroid and either a non-pyrethroid compound e.g. synergists (piperonyl butoxide), pyriproxyfen are under investigation. Most recently, some of these nets received conditional endorsement from WHO to be used in areas reporting moderate insecticide resistance to pyrethroids…..” 

Comment: Lines 152-153 – how was the relative contribution of resistance mechanisms determined, to be able to state that resistance was ‘mainly mediated’ by P450s?

 Responses: This study did not determine the relative contribution of the resistance mechanisms. This colony had already been characterized and the mechanisms of resistance reported by Machani et al 2020. We have cited Machani et al 2020 in the methodology section as REF 29. Briefly, the authors investigated the involvement of the two mechanisms of resistance (kdr and metabolic resistance) through genotyping, enzyme quantification assays and the use of synergist assays (PBO that inhibits the specific activity of p450 monooxygenases). The authors observed partial restoration of pyrethroid susceptibility following synergist pre-exposure suggesting a role of mixed-function oxidases (P450s). 

Comment: Lines 199-201 – this section is not very clear, and it would be good to expand it to describe the behaviour of mosquitoes in more detail.

 Response: We have revised this section from Line 200-205 and now it reads “The working principle of the Mbita bednet trap is that host-seeking mosquitoes will respond to convective plumes, together with the accompanying body odor and exhaled breath from the human bait sleeping under the trap...”

Comment: What is the ‘green dye’ that was used to mark mosquitoes? Fluorescent dust? Give details, with supplier.

 Response: This section has been revised and more details are provided in Line 215-220. The green dye was Fluorescent powder. “The two strains were color-marked with either a green or pink fluorescent powder [FTX Series, Astral Pink; Swada (London) Ltd, London, U.K.] to distinguish them after simultaneous release into the semi-field environment. The powder was applied by filling a syringe (0.5 ml with 0.6 × 25 mm needle) with fluorescent powder. The syringe was held through the gauze at the top of the cup and in one gentle push, the powder was blown out of the syringe. This created a cloud of powder inside the cup with the mosquitoes.”

Comment: Lines 221-222 – it is not clear to me how using F1 wild mosquitoes ‘validates’ these behaviours. Please explain this more clearly. Response: The resistant and susceptible strains used in this work had been colonized for 6 and 14 generations, respectively. The field population was used to validate the observed behaviors between these two strains because of any changes in behavior that may have resulted from colonization.

Comment: Statistical analysis section – would be clearer if it was divided into separate paragraphs. Response: We have revised this section and now we have three separate paragraphs for clarity.

Comment: Line 285 – were these 2,000 F1 mosquitoes included in the 12,000 mentioned in the previous paragraph? Lines 288-291 – should be a separate paragraph Response: The 12,000 mosquitoes released were exclusive of the 2000 F1 mosquitoes.

Comment: Lines 288-291 – should be a separate paragraph Response: We have revised this section accordingly.

Comment: The figures are nice and clear, and I appreciate the use of photos and the labelled diagram of the Mbita bednet trap. I would like to suggest a figure is added, perhaps in place of Figures 3 and 4 – a stacked bar graph, to show the fate of all released mosquitoes of each strain with treated and untreated traps, would show all data from the experiment clearly in one place for easy comparison. Responses: While we appreciate the reviewer’s comments we note that stacked column charts work well when the focus of the chart is to compare the totals and one part of the totals. We strongly feel that doing as suggested by the reviewer, will shift the focus of the manuscript away from the core objectives of the study. We have therefore skipped this comment.

Comment: The first paragraph of the Discussion is quite out of date. Reference 5 is from 2011, and since then there has been a fair amount of research into the effects of insecticide resistance both on the impact of ITNs and on the sub-lethal effects of pyrethroid exposure on resistant mosquitoes, by the same authors and others. It is true that there has been less investigation of the effects on behaviour of mosquitoes, see Review and Meta-Analysis of the Evidence for Choosing between Specific Pyrethroids for Programmatic Purposes by Lissenden et al 2021. However, Phillip McCall at the Liverpool School of Tropical Medicine has done some work on

behavioural responses that deserves mention here, albeit in a lab setting. The final sentence of this paragraph is an important new observation.

 Response: The first paragraph has been revised and we have cited Philip McCall studies on behavior (Hughes et al 2020). We have also replaced the 2011 reference with one from 2016 ( Ranson and Lissenden, 2016). These are on line 349– 351 

Comment: Lines 345-361 – are the authors able to comment on how these two effects (more exit of susceptible mosquitoes v more biting by resistant mosquitoes) might balance against each other in their effect on malaria transmission and/or efficacy of ITNs? At least they could comment that the prevalence and intensity of

resistance in a given area would affect this balance at a local level. Response: We have added a statement on Line 371- 373 that reads “The current findings emphasize the need for continuous monitoring and designing of novel resistance management strategies as the prevalence and intensity of resistance at different locations may have an effect on malaria transmission.” We have also cited Kleinschmidt et al 2018 on the implication of insecticide resistance for malaria vector control.

Comment: Lines 362-364 – the authors describe ITN’s effect giving personal protection, but the other way that ITNs work is to provide community protection by killing mosquitoes, and this should be mentioned. Response: This comment has been addressed and we have included the killing effect as one ITN protection mechanism. Line 391.

Comment: Lines 382-384 – the fact that the ‘susceptible’ strain carried the kdr mutation is an interesting observation, suggesting that this resistance mechanism is not involved in altered behaviour in resistant mosquitoes, but that others might be (such as point-mutations mentioned in the previous paragraph). It might be

interesting to expand this sentence to discuss this. Response: We have provided more information on how other point mutations i.e 1014F have been associated with behavioral costs from Line 397-400 “Studies carried out by Diop et al. [51] in the laboratory on host-seeking behavior of mosquitoes in the presence of damaged treated nets using a wind tunnel, observed increased performance of resistant females with 1014F kdr mutations compared to susceptible counterparts”

Comment: In the Conclusion the effect of outdoor biting is discussed, but there will also, I presume, be more indoor biting by resistant mosquitoes which are not deterred from blood feeding. We have revised the conclusion section by including the impact of indoor biting resistant mosquitoes. Line 429-431 “…This might be a reason for continued malaria transmission indoors in areas with high pyrethroid resistance despite the scaling up of vector control interventions and increased outdoor malaria transmission in sub-Saharan Africa.”

Comment: The manuscript needs a careful edit to remove typos, formatting errors, and inaccurate wording. Response: We have revised and formatted the manuscript carefully and addressed all the typos.

---

## [Editor Report · Decision Letter 1]

21 Mar 2022

Behavioral responses of pyrethroid resistant and susceptible Anopheles gambiae mosquitoes to insecticide treated bed net

PONE-D-21-36181R1

Dear Dr. Afrane,

We’re pleased to inform you that your manuscript has been judged scientifically suitable for publication and will be formally accepted for publication once it meets all outstanding technical requirements.

Kind regards,

Pedro L. Oliveira

Academic Editor

PLOS ONE
---

## [Editor Report · Acceptance letter]

28 Mar 2022

PONE-D-21-36181R1 

Behavioral responses of pyrethroid resistant and susceptible *Anopheles gambiae* mosquitoes to insecticide treated bed net 

Dear Dr. Afrane:

I'm pleased to inform you that your manuscript has been deemed suitable for publication in PLOS ONE. Congratulations! Your manuscript is now with our production department. 

Kind regards, 

on behalf of

Dr. Pedro L. Oliveira 

Academic Editor

PLOS ONE